# Comparison of Fatigue, Quality of Life, Turnover Intention, and Safety Incident Frequency between 2-Shift and 3-Shift Korean Nurses

**DOI:** 10.3390/ijerph18157953

**Published:** 2021-07-27

**Authors:** Jeonghee Hong, Misoon Kim, Eunyoung E. Suh, Sangwoon Cho, Soyoung Jang

**Affiliations:** 1Department of Nursing, Samsung Medical Center, 81 Irwon-ro, Gangnam-gu, Seoul 06351, Korea; jhee.hong@samsung.com (J.H.); ms0622.kim@samsung.com (M.K.); sw337.cho@samsung.com (S.C.); 2Center for Human-Caring Nurse Leaders for the Future by Brain Korea 21 (BK 21) Four Project, Research Institute of Nursing Science, College of Nursing, Seoul National University, 103 Daehak-ro, Jongno-gu, Seoul 03080, Korea; 3College of Nursing, Seoul National University, 103 Daehak-ro, Jongno-gu, Seoul 03080, Korea; jsy8824@snu.ac.kr

**Keywords:** 2-shift work, nurses, fatigue, quality of life, job satisfaction

## Abstract

This study aimed to compare the fatigue, quality of life, turnover intention, and safety incident frequency between 2- and 3-shift nurses, and analyze their perceptions of the 2-shift system. Participants were 227 nurses working for one year or more in a tertiary hospital in Seoul, South Korea (113 were 2-shift nurses for two months or longer, and 114 were 3-shift nurses with no experience of 2-shift work). The Occupational Fatigue Exhaustion Recovery Scale (OFER) and Quality of Life Scale were used. Turnover intention, safety incident frequency, and perceptions of the 2-shift system were surveyed by questionnaires developed by the researchers. Results showed that 2-shift nurses had lower chronic fatigue (t = −2.38, *p* = 0.018) and higher recovery between shifts (t = 3.90, *p* < 0.001) and quality of life scores than 3-shift nurses (t = 3.69, *p* < 0.001). There were no significant differences for turnover intention (t = −1.48, *p* = 0.140), frequency of needlestick accidents (t = 0.30, *p* = 0.763), medication errors (t = −1.46, *p* = 0.146), or near-miss medication errors (t = 0.78, *p* = 0.437). Two-shift nurses found it easier to secure rest and personal leisure time, and their shift system was shown to improve work satisfaction by increasing the continuity of care. Additional research is necessary to examine how nurses’ health status and emotional satisfaction vary by shift type.

## 1. Introduction

Nursing is a job that requires 24/7 care delivery, and is an appropriate representative of shift-based professions. In South Korea, a survey across 173 hospitals with 150 beds or more revealed that about 91% of all nurses were shift workers, with 89.4% being 3-shift nurses [1]. These hospitals, as members of the Hospital Nurses’ Association in South Korea, are medical institutions that are committed to improving nurses’ work environments. Nurses’ health problems caused by shift work affect not only the nurses’ quality of life, but also care delivery quality, hence being an important factor affecting turnover rate in this profession, and constituting a serious national and international problem [2]. Moreover, considering the recent and accelerated changes in the healthcare industry and the increasing demand for nursing personnel, past research has emphasized the need for improvements in expertise for various types of nursing work [3].

Shift work has been shown to cause circadian rhythm disorders, and as a result of insufficient sleep, shift workers tend to experience daily life disorders that can lead to chronic diseases [4]. Particularly, nurses’ work is more difficult than that of other shift workers because of strong physical intensity, high medical difficulty, and tension throughout their work hours. Nurses’ work was shown to lead to higher fatigue, which, in turn, has a negative effect on job satisfaction and quality of life [5]. Moreover, nurses’ fatigue has been reported to increase the risk of safety incidents in hospitals, such as medication errors, patient identification errors, care performance decline, and needlestick accidents during the shift [6]. Another study showed, similarly, that nurses’ accumulated fatigue eventually causes health problems, decreases job satisfaction and quality of life, and is an important cause of job turnover [7]. A recent survey of South Korean nurses showed that about 10% of those working in a 3-shift system listed health problems as the main cause of turnover [8]. Thus, to ensure a stable maintenance of nursing personnel, it may be necessary to study work types, provide safe nursing care, and introduce various types of shifts; these procedures can facilitate such maintenance.

In the United States of America, nurses can choose between 8-, 10-, and 12-h shifts as they desire; among developed countries in Europe, working hours are operationalized based on flextime, meaning that nurses choose their work schedule [9]. To increase the satisfaction of nurses, Japan has introduced various systems, including the daytime fixed-work system, nighttime fixed-work system, flextime system, work-sharing system (i.e., in which one’s work is distributed between two people), and a staggering commute system (i.e., in which working hours can be set freely) [10].

Importantly, over the past 30 years, the 2-shift system (i.e., divided into two shifts of 12 h) has been adopted and implemented in many hospitals; this is particularly true in the United Kingdom and the United States of America, and many nurses and medical institutions have said that they prefer it [11]. Regarding nurses’ practical work, the 2-shift system has the advantages of improving the continuity of care, effective communication with medical staff, and social activities [12]. Regarding nurses’ personal life, it increases the number of holidays, reduces commute time, and improves work–life balance and life flexibility [13,14]. Meanwhile, it positively affects nurses’ health, patient safety, and quality of nursing care [13,15,16].

Nevertheless, this 12-h, 2-shift system has also evoked some concerns in the existing literature; for instance, it may decrease nurses’ concentration owing to accumulated working hours, and increase patient safety incidents owing to drowsiness in nurses [17]. Moreover, a study has described a relationship between nurses working 12 h or more, overtime work, and the risk of patient safety incidents [17]. Concomitantly, studies have reported that 3-shift nurses may incur insufficient physical recovery owing to the 3-shift system, and such insufficient recovery has been associated with higher patient safety incident frequency compared with that in the 2-shift system [18]. Accordingly, this last cited study remarked on the need for more analyses of the association between patient safety incidents and working hours.

According to a review study on shift work by Ferguson and Dawon [19], many studies provide evidence that the 2-shift system is better for sleep quality and work satisfaction, while the 3-shift system is better for patient safety; the review also remarks the lack of nursing research on the effect of the 2-shift system on patient safety or nurses’ wellbeing [19]. In South Korea, specific departments of some hospitals tried to introduce this 2-shift system for the first time, and a comparative study was produced on the topic; by analyzing nurses in the pediatric intensive care unit of a hospital, it was shown that the 2-shift nurses had improved job satisfaction and quality of life, although there was no difference in safety incident frequency [20]. Meanwhile, a South Korean study conducted in an intensive care unit and comparing 2- and 3-shift nurses showed no significant differences in fatigue and quality of life; still, 2-shift nurses’ sleep quality and work satisfaction were better than those of 3-shift nurses [21].

Hence, the above literature review shows that nurses’ shift type is an important factor affecting their physical and mental health status, work–life balance, and quality of life. However, most nurses in South Korea still work in a 3-shift system, and the 2-shift system is only partially applied in a small number of hospitals, so the application of various work types and related studies is limited. In addition, previous studies were conducted on nurses in a specific ward, and studies on both general wards and intensive care units were extremely rare. This study aimed, accordingly, (1) to compare fatigue, quality of life, turnover intention, and safety incident frequency between 2- and 3-shift South Korean nurses, as well as (2) to understand 2-shift nurses’ perceptions of the 2-shift system, and (3) compare satisfaction with current shift type among the 2- and 3-shift nurses. The results of this study are intended to provide evidence for the necessity of improving nurses’ work systems, which may be done by introducing effective ones.

## 2. Materials and Methods

### 2.1. Study Design

This study used a descriptive survey design. Study variables include nurses’ fatigue, quality of life, turnover intention, safety incident frequency, and nurses’ perception of the 2-shift system and satisfaction with their current shift type.

### 2.2. Sample

Study participants were nurses working at a tertiary hospital located in Seoul, South Korea; they were selected from 49 departments, including 38 general wards and 11 intensive care units. The selection criteria were as follows: nursing work experience of one year or more; for 2-shift nurses, they had to be working in a 2-shift system for two months or longer; and for 3-shift nurses, they had to be working in a 3-shift system and not have had experience working in a 2-shift system. Participants were selected in such a way as to ensure that they had experience working in, for example, a 2-shift system for at least 2 months in the same department. The 2-shift nurses were recruited first, and then the 3-shift nurses were recruited according to the proportion of ICU and general ward nurses. The other demographic or work-related characteristics were not matched.

Using G*power, version 3.1(Heinrich Heine Universität Düsselforf, Schwaderer, Germany), sample power was calculated [22]; for conducting independent *t*-tests and two-sided tests with a medium effect size at 0.5, significance level at 0.05, and power at 0.95, a sample of 105 participants was required. While considering a 15% dropout rate, a total of 252 participants were recruited, with 126 participants in each group of 2- and 3-shift nurses.

### 2.3. Data Collection

Data collection was conducted after obtaining approval from the institutional review board of the hospital, as well as after explaining the study’s aims and procedures to study participants and having them sign the consent form. Data were collected using a self-reported questionnaire, which was applied from 29 June 2017 to 4 August 2017. In total, 252 questionnaires were distributed, and a total of 227 (113 from 2-shift nurses and 114 from 3-shift nurses) were used for analysis; 23 questionnaires that could not be retrieved within the survey period, and 2 questionnaires with too many omissions, were excluded.

### 2.4. Measurements

The questionnaire used in this study comprised 94 questions; 8 regarded demographic characteristics, 15 for fatigue, 28 for quality of life, 11 for turnover intentions, 3 for safety incident frequency, 5 for reasons to choose 2-shift system, 1 for willingness to continue 2-shift system, 22 for perceptions about the 2-shift system, and 1 for satisfaction with shift type.

#### 2.4.1. Demographic Characteristics

These questions comprised items on age, sex, marital status, number of children, total work experience, shift work experience, 2-shift work experience, and department.

#### 2.4.2. Fatigue

After acquiring the author’s authorization, the official Korean version of the Occupational Fatigue Exhaustion Recovery Scale, developed by Winwood [5], was used to assess fatigue. It comprises 15 questions: 5 for chronic fatigue, 5 for acute fatigue, and 5 for recovery between shifts. A 7-point Likert scale was used, ranging from 0 (extremely negative) to 6 (extremely positive), with which the scores ranged from 0 to 42. The total scores were converted into a score out of 100 and used for analysis. In this study, the Cronbach’s α was 0.86 for acute fatigue, 0.70 for chronic fatigue, and 0.81 for recovery between shifts.

#### 2.4.3. Quality of Life

A tool developed by Park et al. [23] was used to assess quality of life. It comprises 28 questions: 5 for self-esteem, 8 for working life, 5 for leisure activities, 6 for emotional status, 2 for physical status, and 2 for family relationship. Subjects responded on a 5-point Likert scale, ranging from 1 (not at all) to 5 (strongly agree); the higher the score, the higher the quality of life. Cronbach’s α in Park’s study was 0.895, and Cronbach’s α in this study was 0.90. Cronbach’s α in the subscales was 0.74 for self-esteem, 0.78 for working life, 0.88 for leisure activity, 0.81 for emotional status, −0.02 for physical status, and 0.76 for family relationship. Cronbach’s α of the physical status had a negative value because the number of items was two, but there was no problem in reliability. The correlation between the subscales of quality of life was calculated (Table 1).

#### 2.4.4. Turnover Intention

A tool developed by Kim [24] was used to assess turnover intention. It comprises 11 questions: 7 on turnover intention, and 4 on the reasons for job change. Subjects responded on a 5-point Likert scale, ranging from 1 (not at all) to 5 (strongly agree), with which the total scores ranged from 11 to 55; the higher the score, the higher the turnover intention. Cronbach’s α in Kim’s study was 0.88, and Cronbach’s α in this study was 0.81.

#### 2.4.5. Safety Incident Frequency

Three questions were used to assess safety incident frequency. These asked about the presence and frequency of needlestick accidents in the previous two months, the presence and frequency of medication errors leading to harmful and harmless consequences, and the presence and frequency of near-miss medication errors. The respondents were asked to report frequency of needlestick accident, medication errors, and near-miss medication errors by numbers, and the mean of each frequency was analyzed.

#### 2.4.6. Reasons to Choose a 2-Shift System (Only for 2-Shift Nurses)

Reasons to choose a 2-shift system were queried using 5 item-questionnaire. The items were developed by expert nurses based on their clinical experience. The primary items were reviewed by another expert group to validate their contents. The five reasons to choose a 2-shift system included for sufficient rest, increased time for self-development and hobbies, childrearing, reduced burden of commuting, other. Multiple responses were allowed, and the frequency and percentiles were analyzed.

#### 2.4.7. Willingness to Continue 2-Shift System (Only for 2-Shift Nurses)

The willingness to continue working a 2-shift system for the 2-shift nurses was queried by one dichotomous question with a yes or no response. The frequency and percentiles were analyzed.

#### 2.4.8. Perceptions of the 2-Shift System by 2-Shift Nurses

Participants’ perceptions of the advantages and disadvantages of the 2-shift system were measured by a 22-item questionnaire that was developed by the researcher. The items were developed primarily by the researcher based on three individual interviews with 2-shift nurses. The primary items were reviewed by a group of nursing experts composed of a nursing professor and two clinical nurses, to ensure face validity. The 22 items were divided into personal life and nursing practice, with 12 and 10 questions, respectively. Multiple responses were allowed, so all options with which participants agreed could be described, and the results were analyzed by frequency sequence.

#### 2.4.9. Satisfaction with Shift Type

The degree of satisfaction with current shift type (i.e., either 2- or 3-shift system) was measured using a 5-point Likert scale item, ranging from 1 (extremely dissatisfied) to 5 (extremely satisfied). The higher the score, the higher the satisfaction with shift type. The scores were analyzed by frequency and percentage.

### 2.5. Statistical Analysis

SPSS 22.0(IBM Corp., Armonk, NY, USA) was used for data analysis; participants’ demographic characteristics were analyzed by frequency, percentage, mean, and standard deviation. Using the chi-squared test, Fisher’s exact test, and Student’s *t*-test, comparisons were made for interest variables between the groups of 2- and 3-shift nurses; specifically, fatigue, quality of life, turnover intention, and safety incident frequency were analyzed by mean and standard deviation, while the perception of shift type was expressed by frequency and percentage. The differences in fatigue, quality of life, turnover intention, safety incident frequency, and satisfaction of shift type were analyzed for both groups using a *t*-test.

### 2.6. Ethical Considerations

This study was conducted with the approval of the institutional review board of the hospital (IRB No: 2017-06-084-001). At data collection, participants were told that they could withdraw from the study at any time without penalty, and that they could request that their research data be discarded. To prevent exposure of personal information, the researcher in charge collected data using an anonymous questionnaire, and questionnaires were sealed by study participants before delivery to the researcher; this process was done using a separate collection box to ensure anonymity. Study participants were informed that the collected data would be used for research purposes only, and that it would be stored for 3 years after study completion; in accordance with the Bioethics and Safety Act, after this period, the data would be discarded.

## 3. Results

### 3.1. Demographic Characteristics of 2- and 3-Shift Nurses

Participants’ mean age was 28.42 (±4.17) years for 2-shift nurses and 28.49 (±4.18) years for 3-shift nurses; hence, it was homogenous between the groups (t = −0.14, *p* = 0.892). Regarding marital status, 77.9% of the 2-shift nurses and 72.8% of the 3-shift nurses were unmarried (χ^2^ = 0.78, *p* = 0.376). Regarding the number of children, 85% of the 2-shift nurses and 85.1% of the 3-shift nurses had no children, showing no significant difference between the groups (χ^2^ = 0.00, *p* = 0.978). Work experience was homogenous among the groups (t = −0.24, *p* = 0.812), with a mean of 5.48 (±4.15) years for 2-shift nurses and 5.62 (±4.22) years for 3-shift nurses.

Further, there was no significant difference between groups for shift work experience (t = −0.33, *p* = 0.743), with 5.23 (±3.80) years for 2-shift nurses and 5.39 (±3.93) years for 3-shift nurses. Meanwhile, the average duration of 2-shift nurses’ experience in the 2-shift system was 3.24 (±1.44) months. There was no significant difference between groups (χ^2^ = 0.00, *p* = 0.954) regarding department, with 56.6% of the 2-shift nurses working in general wards, 43.4% of the 2-shift nurses in intensive care units, 57% of the 3-shift nurses in general wards, and 43% of the 3-shift nurses in intensive care units. The preliminary homogeneity test confirmed group homogeneity; there was no statistically significant difference between the groups in any variable (Table 2).

### 3.2. Fatigue, Quality of Life, Turnover Intention, and Safety Incident Frequency in 2- and 3-Shift Nurses

The correlations among the research variables between the 2-shift and 3-shift nurses were calculated (Table 3). The mean chronic fatigue score was lower for 2-shift nurses (62.12 (±15.55) points) than for 3-shift nurses (66.87 (±14.45) points; t = −2.38, *p* = 0.018). There was no significant difference regarding mean acute fatigue scores (t = −0.86, *p* = 0.390; 2-shift nurses: 67.61 (±17.35); 3-shift nurses: 69.65 (±18.27)). The mean recovery between shifts score was significantly higher in 2-shift nurses (47.58 (±18.65) points) compared with 3-shift nurses (38.22 (±17.53) points; t = 3.90, *p* < 0.001).

The mean quality of life score was significantly higher in 2-shift nurses (3.02 (±0.47) points) compared with in 3-shift nurses (2.79 (±0.47) points; t = 3.69, *p* < 0.001). Regarding quality of life subscales, there were no significant differences between the two groups regarding mean scores for self-esteem (t = 0.94, *p* = 0.348), emotional status (t = 1.53, *p* = 0.127), physical status (t = 0.46, *p* = 0.643), or family relationship (t = 1.95, *p* = 0.053). However, the mean score for work life was significantly higher for 2-shift nurses (2.77 (±0.58) points) compared with 3-shift nurses (2.41 (±2.57) points; t = 4.63, *p* < 0.001); the mean score for leisure activities was also significantly higher for 2-shift nurses (3.05 (±0.80) points) than for 3-shift nurses (2.53 (±0.79) points; t = 5.00, *p* < 0.001).

The mean turnover intention score was non-significantly different between the two groups (2-shift nurses: 3.28 (±0.55); 3-shift nurses: 3.40 (±0.65); t = −1.48, *p* = 0.140). The differences regarding scores for the frequency of needlestick accidents (t = 0.30, *p* = 0.763), medication errors (t = −1.46, *p* = 0.146), and near-miss medication errors (t = 0.78, *p* = 0.437) between the two groups were also non-significant (Table 4).

### 3.3. Perception of the 2-Shift System by 2-Shift Nurses

#### 3.3.1. Reasons to Choose and Continue to do the 2-Shift System

This study applied a questionnaire to assess the reasons to choose a 2-shift system among 2-shift nurses (*n* = 113). In the study sample, 58.9% of the 2-shift nurses selected “sufficient rest”, followed by “increased time for self-development and hobbies” at 12.5%, “childrearing” at 9.8%, and “reduced burden of commuting” at 8.9%. Furthermore, the willingness of 2-shift nurses to continue working in the 2-shift system was surveyed. The results showed that most 2-shift nurses (98.2%) answered that they were willing to continue working in the 2-shift system.3.3.2. Perception of the 2-Shift System by 2-Shift Nurses.

The advantages and disadvantages of the personal life and nursing practice aspects of the 2-shift system for 2-shift nurses were investigated, allowing for multiple responses (Table 5). Regarding personal life advantages, 90.3% of the 2-shift nurses selected “increased rest time”, followed by “increased self-development and leisure time” at 55.8% and “reduced emotional exhaustion” at 31.9%. Regarding personal life disadvantages, 82.3% of the 2-shift nurses selected “increased physical fatigue”, followed by “reduced concentration owing to long work hours” at 61.1%, “increased drowsiness” (31.0%), “increased emotional exhaustion” (27.4%), and “difficulty in participating in education and department meetings” (25.7%).

Regarding nursing practice advantages, 86.7% of the 2-shift nurses answered “increased continuity of care owing to reduced shift change frequency”, followed by “secured time for nursing care planning and delivery” at 60.2%, and “improved teamwork owing to fewer changes in team members when changing shifts” at 50.4%. Regarding nursing practice disadvantages, 55.8% of the 2-shift nurses chose “difficulty in recognizing guidelines changes”, followed by “increased risk of patient safety incident” and “difficulty in patient assessment owing to long work hours in consecutive holidays”.

Another questionnaire was applied for 2-shift nurses to analyze the rate of change in the workload compared with the rate of the increase in work hours in the 2-shift system; its results showed that 51.8% of the 2-shift nurses felt that the workload increased at the same rate as the work hours increased. However, 33.6% responded that the rate of workload increase was higher than that of work hours. As the result of another questionnaire on the effects of the 2-shift system on reduction of resignation and increased retention of 2-shift nurses, 49.6% answered that it was “helpful”, 25.7% answered “average”, and 15.0% answered that it was “not helpful”.

#### 3.3.2. Comparison of Satisfaction with Current Shift Type

In total, 53.1% of the 2-shift nurses answered that they were “very satisfied” or “satisfied” with their current shift system, while 52.7% of the 3-shift nurses answered “dissatisfied” or “very dissatisfied” with their current shift system. There was a significant difference between the two groups (χ^2^ = 61.37, *p* < 0.001; Table 6).

## 4. Discussion

The working hours and conditions of nurses—professionals who deliver care for patients 24/7—are important influencing factors of patient treatment outcomes and nursing workforce retention. This study was conducted in an effort to understand the effectiveness and feasibility of the 2-shift system as a flexible shift system. This study confirmed that there were significant differences in chronic fatigue, recovery between shifts, quality of life, and satisfaction with current shift type between 2- and 3-shift nurses. Furthermore, by assessing nurses’ perceptions about the 2-shift work system, this study attempted to present an evaluation of—and areas of improvement for—the 2-shift system.

The study findings showed that 2-shift nurses had lower chronic fatigue than 3-shift nurses, but showed no significant difference in acute fatigue. While there are past studies that corroborate these findings, and show that the long work hours of 12-h shifts increase nurses’ fatigue [25], there are also studies that show that they reduce fatigue [26], or even that there is no difference [27]. Although the perception of fatigue according to shift type was not consistent in the current study, it has already been reported that nurses’ fatigue owing to shift work and overtime work is high [28]. Furthermore, a study showed that a short recovery time between workdays causes excessive fatigue, drowsiness, and depression [29]; these causal relationships can be inferred as the potential motives behind the lower chronic fatigue of 2-shift nurses than of 3-shift nurses in the current study’s findings. Specifically, 2-shift nurses may have had better opportunities to complete a long recovery time between shifts. Thus, to reduce nurses’ fatigue owing to long work hours, nursing managers should provide active management that assures the allocation of sufficient non-workdays, without exceeding the number of workdays.

This study confirmed that quality of life scores were significantly higher for 2- than for 3-shift nurses. Among the quality of life subscales, scores for work life and leisure activities were higher for 2- than for 3-shift nurses; this is consistent with a study showing positive results in job satisfaction and flexibility in personal life for 12-h shift workers, except for the area of family relationships [14]. Moreover, satisfaction with the 2-shift system has been associated with more vacation days, opportunities for individual activities, and a better work–life balance [30]; specifically, this cited study showed, similarly to the current study, that having greater flexibility to perform leisure activities owing to an increased number of holidays influenced nurses’ quality of life. Moreover, although both groups of 2- and 3-shift nurses showed their highest quality of life scores for the subscale of self-esteem, the difference between the scores for this subscale was non-significant between the groups.

Additionally, the study’s findings show that both groups had low physical status scores; this may be owing to shift-work-related fatigue, which may be generated not only by long working hours but also by the severity of cases in tertiary hospitals—the participating institutions in the current manuscript. However, there are only a few studies on the quality of life of South Korean nurses according to shift work patterns. Thus, various studies are thought to be needed, including intervention studies that can improve the quality of life for shift workers; furthermore, it is necessary to develop methods to enhance the work–life balance for shift workers.

In this study, there was no significant difference between the two groups in the frequency of safety incidents for the 2-shift and 3-shift nurses. This is consistent with the previous study of Lim [20], and it is thought that a 2-shift system reduces the frequency of handover, reduces the complexity of communication between medical staff, and improves nursing continuity. In addition, according to the study, the occurrence of patient safety incidents was related to long working hours of 13 h or more; thus, overtime should not occur in the 12-h shift [17].

Our study included a turnover intention variable, which is rarely found in previous studies. The results of this study showed that the turnover intention of 3-shift nurses was slightly higher than that of 2-shift nurses, but there was no significant difference. Another study found that nurses who worked 12 h or more had a higher turnover intention than nurses who worked 8 h or less [31]. Therefore, there is a need for continuous efforts by nursing organizations to reduce additional overtime after 2-shift work.

In this study, nurses’ reasons to participate in and continue the 2-shift system were surveyed; numerous participants answered that they chose this system owing to it providing them with sufficient rest, self-development, and time for performing hobbies. This is believed to be related to their expectations regarding the nature of the 2-shift system; it is indeed designed in a way that allows for nurses to have multiple long holidays. These may help them to allocate time for practicing self-development, and to improve their work–life balance. This is in the same context as the study that 3 or more days off in a week provide sufficient time for leisure activities and flexibility in work and personal life [32]. Moreover, 98.2% of the 2-shift nurses answered that they were willing to continue to work the 2-shift system, indicating that most were satisfied with their current shift work system [11,19]. However, since this study was conducted at the early stage of the introduction of the 2-shift system, there is the possibility that it may have a positive effect on job satisfaction.

This study also examined the advantages and disadvantages of personal life and nursing practice aspects of the 2-shift system as perceived by 2-shift nurses. Regarding the personal advantages of the 2-shift system, most nurses regarded “increased rest time” and “increased self-development and leisure time” as such. This result is consistent with the aforementioned discussions surrounding increased quality of life and motivation for participation in nurses under the 2-shift system [30]. Regarding the disadvantages, “increased physical fatigue” and “decreased concentration owing to long work hours” were the most selected by 2-shift nurses. A study showed that it is important to provide sufficient rest time to prevent specific factors (i.e., fatigue, inter-shift recovery) from affecting nursing performance [24]. Furthermore, in medical institutions that implement the 2-shift system, a study described that policies such as setting a relevant number of workdays and limiting overtime work should be implemented, as they may help to prevent increased fatigue owing to long working hours and safety incidents caused by poor concentration [33].

Regarding nursing practice advantages of the 2-shift system, most nurses regarded “increased continuity of care” and “secured time for nursing care planning and delivery” as such. Regarding the nursing practice disadvantages, most selected “difficulty in recognizing guidelines changes” and an “increased risk of patient safety incident”. These results can be interpreted considering the understanding that the 12-h shift that characterizes the 2-shift system provides sufficient time for nursing care delivery and increases the continuity of care by reducing the number of handovers; this result and these assumptions are consistent with prior research [20]. However, it may indeed be difficult for 2-shift nurses to quickly learn their new guidelines upon returning to work after a holiday. Accordingly, it may help to prevent work errors if hospital and nursing managers devote their attention to this matter, and motivate communication between colleagues to solve knowledge gaps regarding hospital guidelines. This is in the same context as the study that emphasized the importance of communication in performing safe patient care in an interdependent organizational culture [34].

Finally, there was a significant difference in satisfaction with shift system between the 2- and 3-shift nurses. This is consistent with a previous study that reported that nurses working 12-h shifts showed significantly higher satisfaction with their schedule than those working 8-h shifts [15], which is an important factor affecting job satisfaction. However, the current study enrolled and analyzed data from nursing staff of tertiary hospitals; since the participants in the cited study differ from those in the current sample regarding the size of the hospitals in which they work, it may be that satisfaction with shift system differs by hospital size. Hence, future research is warranted to conduct thorough examinations of this potential relationship. Moreover, the current findings should be interpreted with care.

Although this study compared the perceptions of 2- and 3-shift nurses from multiple perspectives, it still entails methodological limitations. First, since the study targeted only shift-working nurses at a single tertiary general hospital in Seoul, South Korea, caution should be exercised upon generalizing the results to nurses at medical institutions other than tertiary hospitals in that specific location. Second, the long-term effects of the 2-shift system could not be investigated because this study was conducted five months after the introduction of the 2-shift system in the participating hospital. Third, our study relied on self-reported measures of safety incident frequency instead of hospital system records; however, the judgment of nurses who are aware of the importance of safety incidents in hospitals through education might be reliable.

Based on the study’s findings, two suggestions for research can be put forward: Considering the timing of the introduction of the 2-shift system and the assessments of the current paper, it may be necessary to conduct repeated studies on the matter in the future to re-evaluate the effects of the 2-shift system on nurses. Moreover, to prove the effectiveness of the 2-shift system, future studies should expand the study sample to include medical institutions of various sizes.

It is noteworthy that this study was a comparative analysis of nurses’ various perceptions of the 2-shift system attempted in a tertiary hospital. Above all, it has the strength of increasing objectivity according to the number of similar samples of 2-shift and 3-shift nurses. The maintenance of nursing personnel is an important issue responsible for national health problems. Based on these results, it is expected that a more effective working pattern will be established for the human resource management of nurses in the future. We intend to conduct an expanded study on the effectiveness of work types by working department.

## 5. Conclusions

This study compared the fatigue, quality of life, turnover intention, and safety incident frequency of 2- and 3-shift nurses in tertiary general hospitals. Moreover, perceptions of the 2-shift system by 2-shift nurses were examined from various perspectives. This study demonstrated that the increased rest time, time for self-development and hobby activities, and continuity of care that were ensured by engaging in the 2-shift system improved 2-shift job satisfaction. It was confirmed that 2-shift nurses had lower chronic fatigue and higher quality of life compared with 3-shift nurses, and that they prioritized work–life balance. Nevertheless, there were no differences in safety incident frequency or near-miss medication errors between the 2-shift-and 3-shift nurses.

Therefore, it was confirmed that the 2-shift system positively affects nurses’ job satisfaction in terms of shift system and quality of life; it should thus be expanded and applied as a form of shift work. The maintenance of nursing personnel is an essential resource responsible for national health problems. This study is worthwhile in establishing the basis for human resource management policies by confirming the applicability of the 2-shift system. Future, longitudinal, large-sample studies are warranted, as well as studies that analyze potential institutional improvements for nursing staff management, and that examine and introduce possibly more effective shift systems.

## Figures and Tables

**Table 1 ijerph-18-07953-t001:** Correlations between subscales of quality of life.

Variables	SE	WL	LA	ES	PS	FR
Quality of life						
Self-esteem (SE)	1					
Working life (WL)	0.499 *	1				
Leisure activity (LA)	0.348 *	0.550 *	1			
Emotional status (ES)	0.466 *	0.521 *	0.373 *	1		
Physical status (PS)	0.098	0.212 *	0.384 *	0.129	1	
Family relationship (FR)	0.236 *	0.264 *	0.383 *	0.249 *	0.148 *	1

* *p* < 0.05.

**Table 2 ijerph-18-07953-t002:** Demographic characteristics and homogeneity test for 2- and 3-shift nurses at baseline (*n* = 227).

Variable	Categories	2-Shift (*n* = 113)	3-Shift (*n* = 114)	t or χ2	*p*
		*n*(%) or M ± SD	*n*(%) or M ± SD		
Age (years)		28.42 ± 4.17	28.49 ± 4.18	−0.14	0.892
Sex *	Male	8 (7.1)	2 (1.8)	3.82	0.059
	Female	105 (92.9)	112 (98.2)		
Marital status	Married	25 (22.1)	31 (27.2)	0.78	0.376
	Single	88 (77.9)	83 (72.8)		
Have children	Yes	17 (15.0)	17 (14.9)	0.00	0.978
	No	96 (85.0)	97 (85.1)		
Clinical experience (years)		5.48 ± 4.15	5.62 ± 4.22	−0.24	0.812
Shift work experience (years)		5.23 ± 3.80	5.39 ± 3.93	−0.33	0.743
2-shift work experience (months)		3.24 ± 1.44			
Unit	General	64 (56.6)	65 (57.0)	0.00	0.954
	ICU	49 (43.4)	49 (43.0)		

* Fisher’s exact test.

**Table 3 ijerph-18-07953-t003:** Correlations between fatigue, quality of life, turnover intention, and safety incident frequency between the 2-shift and 3-shift nurses.

	2-Shift Nurses						3-Shift Nurses					
Variables	F1	F2	F3	Q	T	S	F1	F2	F3	Q	T	S
Fatigue												
Chronic fatigue (F1)	1						1					
Acute fatigue (F2)	0.542 *	1					0.652 *	1				
Recovery between shifts (F3)	−0.424 *	−0.500 *	1				−0.514 *	−0.612 *	1			
Quality of life (Q)	−0.687 *	−0.573 *	0.482 *	1			−0.540 *	−0.611 *	0.628 *	1		
Turnover intention (T)	0.525 *	0.297 *	−0.403 *	−0.593 *	1	*	0.490 *	0.390 *	−0.443 *	−0.505 *	1	
Safety incident frequency(S)	−0.213 *	−0.301 *	0.325 *	0.323 *	−0.205 *	1	−0.422 *	−0.355 *	0.387 *	0.567 *	−0.577 *	1

^*^*p* < 0.05; F1: chronic fatigue; F2: acute fatigue; F3: recovery between shifts; Q: quality of life; T: turnover intention; S: safety incident frequency.

**Table 4 ijerph-18-07953-t004:** Distribution for fatigue, quality of life, turnover intention, and safety incident frequency by group (*n* = 227).

Variable	Categories	2-Shift Nurses (*n* = 113)	3-Shift Nurses (*n* = 114)	t	*p*
		M ± SD	M ± SD		
Fatigue					
Chronic fatigue		62.12 ± 15.55	66.87 ± 14.45	−2.38	0.018
Acute fatigue		67.61 ± 17.35	69.65 ± 18.27	−0.86	0.390
Recovery between shifts		47.58 ± 18.65	38.22 ± 17.53	3.90	<0.001 *
Quality of life		3.02 ± 0.47	2.79 ± 0.47	3.69	<0.001 *
Self-esteem		3.64 ± 0.54	3.57 ± 0.54	0.94	0.348
Working life		2.77 ± 0.58	2.41 ± 2.57	4.63	<0.001 *
Leisure activity		3.05 ± 0.80	2.53 ± 0.79	5.00	<0.001 *
Emotional status		2.83 ± 0.75	2.68 ± 0.71	1.53	0.127
Physical status		2.54 ± 0.73	2.50 ± 0.84	0.46	0.643
Family Relationship		3.31 ± 0.90	3.07 ± 0.92	1.95	0.053
Turnover intention		3.28 ± 0.55	3.40 ± 0.65	−1.48	0.140
Safety incident frequency					
Needlestick accident		0.33 ± 0.89	0.29 ± 0.73	0.30	0.763
Medication error		0.04 ± 0.21	0.10 ± 0.33	−1.46	0.146
Near-miss medication error		0.30 ± 0.67	0.23 ± 0.66	0.78	0.437

* *p* < 0.05.

**Table 5 ijerph-18-07953-t005:** Comparison of the advantage and disadvantage scores of the 2-shift system (*n* = 113).

Variables	Categories	*n* (%)
Advantages	Personal life	
	Increased rest time	102 (90.3)
	Increased self-development and leisure time	63 (55.8)
	Reduced emotional exhaustion	36 (31.9)
	Increased participation in child care	14 (12.4)
	Decreased physical fatigue	12 (10.6)
	Other	16 (14.2)
	Nursing practice	
	Increased continuity of care owing to reduced shift change frequency	98 (86.7)
	Secured time for nursing care planning and delivery	68 (60.2)
	Improved teamwork owing to fewer changes in team members when changing shifts	57 (50.4)
	Improved relationship with patient/caregiver	37 (32.7)
	Increased communication with other medical staff	13 (11.5)
	Other	6 (5.3)
Disadvantages	Personal life	
	Increased physical fatigue	93 (82.3)
	Reduced concentration owing to long work hours	69 (61.1)
	Increased drowsiness	35 (31.0)
	Increased emotional exhaustion	31 (27.4)
	Difficulty participating in education and department meetings	29 (25.7)
	Other	5 (4.4)
	Nursing practice	
	Difficulty in recognizing guidelines changes	63 (55.8)
	Increased risk of patient safety incident	34 (30.1)
	Difficulty in patient assessment owing to long work hours in consecutive holidays	20 (17.7)
	Other	11 (9.7)

**Table 6 ijerph-18-07953-t006:** Comparison of participants’ satisfaction with their current shift system (*n* = 225).

Shift System Satisfaction	2-Shift Nurses (*n* = 113)	3-Shift Nurses (*n* = 112) *	χ^2^	*p*
	*n* (%)	*n* (%)		
Very satisfied	10 (8.9)	0 (0.0)	61.37	<0.001
Satisfied	50 (44.2)	11 (9.8)		
Neutral	38 (33.6)	42 (37.5)		
Dissatisfied	13 (11.5)	49 (43.8)		
Very dissatisfied	2 (1.8)	10 (8.9)		

* Excluded 2 non-responses

## Data Availability

Data are available upon request.

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
