# Peer review of "Comparison of Fatigue, Quality of Life, Turnover Intention, and Safety Incident Frequency between 2-Shift and 3-Shift Korean Nurses"

_ijerph, 2021, doi:10.3390/ijerph18157953_

Round 1
Reviewer 1 Report
I appreciate the opportunity to read the manuscript entitled „Comparison of Fatigue, Quality of Life, Turnover Intention, and Safety Incident Frequency between 2-Shift and 3-Shift Korean Nurses“ submitted tot he International Journal of Environmental Research and Public Health. Leveraging two matched samples of nurses from a hospital in South Korea this study compares occupational well-being and self-reported safety performance either working in 2-shoft or a 3-shift system. My sense is that this manuscript has been prepared very conscientiously and that the analyses do make sense. As with any manuscript I identified a couple of issues that should be considered before moving on. I outline the major aspects below and hope my comments will be helpful as you go on with this line of research.
- I think you give a clear review of the evidence on the pros and cons of 2-shift vs. 3-shift systems. I really like your point that there is contradictory evidence on for instance the 2-shift-system on lines 68-77. I think the ambivalent effects of the 2-shift-system really call for more research. However, your study appears rather descriptive to me. Could you think of ways to make a stronger or a more explicit case why a study like yours ist he logically most straightforward step to resolving the inconsistencies?
- From a practical perspective your study appears highly valuable to me, because it may help managers in charge of work design decide on an evidence-based approach which shift-system to prefer from the perspective of human research management. From a more academic perspective, however, I wondered whether you can really draw strong conclusions beyond the specific context studied. When discussing the findings you could refer to previous evidence to argue for the generalizability if you replicate findings and justify why your study is similar to empirical work on shift work in other contexts.
- I am impressed that you obviously managed to recruit two perfectly matched samples. This is a clear strength of your paper, as it helps rule out alternative explanations for the differences found. I must admit that it is hard for me to believe that the perfect matching happened purely by chance. Could you please elaborate on the sampling strategy in the paper? For instance, how did you decide who to invite for participation. Can you provide some context on the population of nurses in the organization studied? Could you also elaborate a bit on the criteria for why a nurse may be working in a 2-shift or a 3-shift system? Are nurses free to choose? From your manuscript; i suspect it is not the case. So, do different departments apply different approaches here?
- I think the T-tests make sense, although a regression-based approach would allow controlling for additional variables beyond type of shift-system. In the theory section you argue that working in intensive care vs. general wards may differ systematically and this may explain the inconsistent prior evidence. I wonder if controling for general ward vs. intensive care or better including this variable as a substantial predictor would help address the role of general ward vs. intensive care. My sense is that if you want to address this issue empirically you should include type of department as a covariate/substantial predictor in your analyses. If you find that different shift-systems work differently across general wards vs. intensive care this would really resolve the inconsistency and it would make use of the rich data you. If you choose to follow this line, a regression approach taking type of shift-system, type of job, etc. as predictors might be the most straightforward option. If you decide to stick tot he T-Tests, you run the risk of providing a primarily descriptive study with unknown generalizability.
- I noticed that you do not provide a correlation table. This is common practice for quantitative studies at least in the social sciences. To get an idea please see Table 2 in Koopman et al. (2016). I find correlation tables extremely important as they they allow including published papers or any kind of report in met-analyses on, say fatigue and tunrover intention. I suggest you report correlations for both sub-samples separately (e.g., below vs. above the diagonal). This way more detailed comparisons beyond the descriptives (mean, SD) of the sub-samples are easy to do for readers.
- In line 151 you report an alpha based on a scale consisting of 28 items. I strongly recommend that you either report reliabilities fort he facets of quality of life or present an alternative measure of reliability such as McDonald’s Omega, because Alpha is misleading with long scales (see Hayes, 2020).
Koopman, J.; Lanaj, K.; Scott, B.A. Integrating the Bright and Dark Sides of OCB: A Daily Investigation of the Benefits and Costs of Helping Others. ACAD MANAGE J 2016, 59, 414–435, doi:10.5465/amj.2014.0262.
Hayes, A.F.; Coutts, J.J. Use Omega Rather than Cronbach’s Alpha for Estimating Reliability. But…. Communication Methods and Measures 2020, 14, 1–24, doi:10.1080/19312458.2020.1718629.
Author Response
"Please see the attachment."
Thank you.

Reviewer 2 Report
The article is well written, but there are some needs for improvement.
1) Clarify how the hospital under study fits into the context of the 173 hospitals described in the introduction. Mention the most relevant characteristics of the hospital under study.
2) It is very necessary to insert an appendix with the survey designed [line 104]. Fatigue (Winwood), Quality of life (Park), Turnover intention (Kim), Safety incident frequency (3 items), Perceptions ... (20-item ¿?), and Satisfaction ... (¿?). Tables 2, 3 and 4 also do not allow us to know the variables studied.
3) Add some citations of studies where G*power is used, what are the benefits of its use?
4) The discussion should add more recent references to highlight the importance and achievements of this article.
5) The data have been collected in 2017 [lines 124-125]. At the level of conclusions, how this study can be projected to the present (under health crisis) and in a future line of research.
Author Response
"Please see the attachment."
Thank you.

Round 2
Reviewer 1 Report
Thank you for giving me the opportunity to read the revised version of the manuscript entitled “Comparison of Fatigue, Quality of Life, Turnover Intention, and Safety Incident Frequency between 2-Shift and 3-Shift Korean Nurses submitted to the International Journal of Environmental Research and Public Health. My overall assessment of the original submission was quite positive, because I think that the design and the analysis is rigorous and makes sense.
- In my last review I therefore focused on conceptual or theoretical issues. For instance, I suggested striking out more clearly the gap in the literature and how you study helps address this question. Although I believe the contribution is clear implicitly, when I look at your study more rigorously the added values appears very limited, because you do not provide specific arguments how study builds on inconsistencies or gaps and you do not explain why the specific context, the specific variables and the specific sample are a perfect fit to address the gaps identified. Could think of ways to elaborate a bit more on what is the scientific value of the study beyond prior research and beyond more or less descriptive results on a specific context.
- In my last review, I suggested to consider regression analysis to address more specific conceptual issues like the distinction between different types of wards. Honestly, your justification for not following this suggestions does not convince me. You refer to future research that might apply superior designs, but the data at hand would permit for more fine grained analyses. I am not quite sure how to proceed here. My sense is that you should either drop the statements about differences across different types or wards, because you do not address this issue. Or you should try to address this issue and consider making this a contribution of the paper. A pragmatic solution might be to report some supplemental analyses on differences across different wards. A very simple solution might be to run separate T-test for subgroups of different wards. If there are no differences in the patterns of results (T-values for comparison of shift-systems across ward types) or if they are negligible you could mention this briefly in the manuscript and you could discuss implications of the finding in the discussion.
- Thank you for adding the correlation table. This is very helpful for me and valuable for readers, I think.
- It seems like my comment on the reliability of the quality of life scale was not as clear as it should be. You mention that quality of life consists of different sub-scales or facets (like self-esteem). To me they seem to be very dissimilar conceptually. Can you please (a) report the reliabilities for quality of life at the facet level (alpha for self-esteem etc.)? Can you please (b) add the quality of life facets in the correlation table? Can you please (c) run the T-tests for the quality of life facets *besides the analyses for the overall quality of life index*? This would provide a more nuanced picture of which facets of quality of life are affected the most by the shift-system. This may even be a contribution in its own right, because you add precision to the study of quality of life.
- I think you have responded to my comment on context a bit superficially. Can you please elaborate a bit more on the context of the sample and jobs? For instance, what kind of patients are the nurses confronted with. What does the job (tasks, schedule, interaction with clients, …) look like in the different types of wards?
- Can you please provide an explanation of what is a tertiary hospital?
- You please elaborate a bit more on the strengths and limitations of the study. For instance, the similarities between the two samples is clearly a strength of your study and you could make this obvious to readers. Concerning limitations, I would prefer to have a bit more elaboration on what are the implications of the limitations. To extent may we trust the findings although study design etc. is imperfect and generalizability is unknown?
As you can see from my comments, I do appreciate the changes you made in response to my comments and those of reviewer 2. However, I expect authors to be a bit more responsive to concrete and manageable suggestions made by reviewers. This means that I expect that authors do not merely state that I made good suggestions but authors prefer to more or less ignore them (anyway). Reviewers may become more rigorous or insist on major changes, once they sense that authors address issues superficially and try to provide quick fixes to rather important (conceptual) issues. My advice would be to work very thoroughly from the beginning.
Author Response
Dear Reviewer,
We appreciate your comments. We revised the manuscript according to your comments. Please see the attached change table. Thank you.
